# Vitamin D and the Risk of Developing Hypertension in the SUN Project: A Prospective Cohort Study

**DOI:** 10.3390/nu16142351

**Published:** 2024-07-20

**Authors:** Ana Valer-Martinez, Maira Bes-Rastrollo, Jose Alfredo Martinez, Miguel Angel Martinez-Gonzalez, Carmen Sayon-Orea

**Affiliations:** 1Department of Preventive Medicine and Public Health, University of Navarra, 31008 Pamplona, Spain; anavalerm@gmail.com (A.V.-M.); mamartinez@unav.es (M.A.M.-G.); msayon@unav.es (C.S.-O.); 2Department of Family Medicine, SALUD Aragon, 50009 Zaragoza, Spain; 3CIBERobn, Instituto de Salud Carlos III, 28029 Madrid, Spain; jalfmtz@unav.es; 4Nutrition Program, IdiSNa, Navarra Institute for Health Research, 31008 Pamplona, Spain; 5Institute IMDEA Food, 28049 Madrid, Spain; 6Department of Nutrition, Food Science and Physiology, Centre for Nutrition Research, University of Navarra, 31008 Pamplona, Spain; 7Harvard T.H. Chan School of Public Health, Boston, MA 02115, USA; 8Navarra Public Health Institute, 31003 Pamplona, Spain

**Keywords:** predicted vitamin D, vitamin D deficiency, hypertension, blood pressure, prospective cohort

## Abstract

Vitamin D deficiency has been associated with a higher risk of multiple diseases, including cardiovascular disorders. The purpose of this study was to examine the potential association between predicted levels of serum 25(OH)D and the risk of new-onset hypertension in a large Mediterranean cohort. A validated 136-item food frequency questionnaire was used as the dietary assessment tool. 25(OH)D serum levels were predicted using a previously validated equation. We performed Cox regression models to analyze the association between predicted serum 25(OH)D and the risk of hypertension, according to quartiles of forecasted vitamin D at baseline, after adjusting for multiple potential confounders. Over a median follow-up of 12.3 years, 2338 new cases of hypertension were identified. The analyses revealed a significant inverse association between predicted serum levels of 25(OH)D at baseline and the risk of hypertension. Individuals in the highest quartile showed a 30% relatively lower risk of hypertension compared to the lowest quartile (hazard ratio (HR): 0.70; 95% confidence interval (CI): 0.60–0.80, *p*-trend < 0.001). The outcomes remained significant after performing sensitivity analyses. The findings suggested that higher levels of forecasted vitamin D are inversely and independently associated with the risk of incident hypertension, implying that vitamin D may offer protective benefits against the disease.

## 1. Introduction

Hypertension, also known as high blood pressure, is the most prevalent cardiovascular disorder in the world [1]. The World Health Organization (WHO) has pointed out that nearly 1.28 billion adults worldwide between the ages of 30 and 79 have been diagnosed with hypertension, two-thirds residing in low-income and middle-income regions [1]. In individuals below 50 years old, hypertension is more commonly found in men. However, the sharper rise in systolic blood pressure (SBP) in women from their third decade of life, particularly after menopause, results in a higher prevalence of high blood pressure among women in groups over 65 years [2,3]. The relevance of hypertension lies in its close association with a higher risk of heart failure, stroke, and the development and progression of chronic kidney disease and coronary artery disease (CAD). Vitamin D deficiency has emerged as a global health concern [4,5,6]. Vitamin D is a fat-soluble vitamin, which is synthesized endogenously via sun exposure to the skin. Its metabolites, cholecalciferol and ergocalciferol, are naturally found in various dietary sources (mainly fatty fish—such as salmon, rainbow trout, or herring—and egg yolks, among others) [4]. Beyond its well-established role for calcium in bone metabolism, 1,25(OH)_2_D (the active form of 25(OH)D) has been implicated in diverse physiological processes, including metabolic disorders [7,8], cardiovascular diseases (CVD) [9], or the modulation of the immune response [10].

Various observational studies conducted with data from the third National Health and Nutrition Examination Survey (NHANES III) concluded that vitamin D deficiency was inversely associated with essential hypertension [11,12]. Although it remains unclear which exact mechanisms underlie this association, several authors have suggested different hypotheses [13,14]. Nevertheless, other randomized clinical trials (RCTs) showed weak evidence supporting the positive impact of vitamin D supplementation on blood pressure [14,15,16,17]. Moreover, results from several published meta-analyses have not been able to reach any common consensus [16,18,19,20,21,22,23]. In fact, some of them proved a beneficial effect of vitamin D supplementation in specific groups, such as older subjects or obese participants [19,20,23]. Given the potential presence of residual confounding, the task of deducing causality or consequence in the association and establishing a consensus based on these findings becomes challenging [22].

As sunlight constitutes a major source of vitamin D and its availability varies widely based on different factors such as latitude, geography, and lifestyle, the applicability of previous research findings to the Spanish population may be limited. Therefore, this study seeks to shed light on the potential link between forecasted serum 25(OH)D and the likelihood of developing hypertension within a sizable healthy Mediterranean cohort followed over an extended period.

## 2. Materials and Methods

### 2.1. Study Population

The SUN project (Seguimiento Universidad de Navarra, Pamplona, Spain) is a dynamic, prospective, and multipurpose cohort, which includes Spanish university graduates. The recruitment of participants started in December 1999 [24]. Data were gathered from participants through validated questionnaires distributed at baseline and every two years. These surveys encompassed information about dietary patterns, lifestyle factors, or the prevalence and incidence of various diseases at both the initial assessment and during subsequent follow-ups, among others. By 31 May 2022, a total of 23,133 individuals completed their initial questionnaire. For this analysis, we excluded 2525 individuals who had previously been diagnosed with hypertension at the time of enrollment. We considered only participants who completed the baseline questionnaire before 31 August 2019 (*n* = 20,384) to warrant at least 2 years and 9 months of follow-up. Additionally, we excluded 1900 individuals with reported daily energy intakes outside the established limits set by Willett (women: <500 kcal/day or >3500 kcal/day; men: <800 kcal/day or >4000 kcal/day) [25]: 757 subjects with prevalent chronic diseases at baseline (diabetes, CVD and cancer) and 1290 participants who were lost to follow-up. Ultimately, our analysis included 16,437 participants and 2338 new cases of hypertension (Figure 1). The overall long-term retention rate in the cohort was 92.7%.

This research followed the principles outlined in the Declaration of Helsinki, and all the procedures were approved by the Research Ethics Committee of the University of Navarra.

### 2.2. Exposure Assessment

A validated predictive model, established through multiple linear regression analyses [26], was used to estimate serum 25(OH)D levels for each participant. This predictive model incorporated several variables: age, sex, vitamin D intake from diet and supplementation, body mass index (BMI) derived from self-reported and validated weight and height (kg/m^2^) [27]; skin reaction to sun exposure (mild or severe reaction); daily time spent walking (minutes per day); and sunlight exposure during summer (hours per day). All these variables were included in the following equation:24.4 + (0.04 × dietary vitamin D) + (−0.01 × age) + (1.37 × sex) + (−0.31 × BMI) + (−3.71 × skin phototype) + (0.03 × walking time) + (0.77 × summer sun exposure) + (0.03 × physical activity)

Dietary data were acquired through a validated semiquantitative food frequency questionnaire (FFQ) covering 136 items [28,29,30]. Vitamin D intake from both dietary sources and supplements was consolidated into a single variable, adjusted for total energy intake (kcal) using the residual method. Summer sun exposure was assessed after calculating a weighted average of the exposure to sunlight each week, during summer months. Physical activity was evaluated through a questionnaire, which were previously validated, in which each physical activity was assigned a weight based on its proportional metabolic equivalents (METs) [31]. Additionally, the duration of each activity was considered to calculate a comprehensive value of MET hours per week for every participant. The methodology of the mentioned predictive model has been thoroughly detailed in previous descriptions [26].

### 2.3. Outcome Assessment

The main outcome of the current analyses was the incidence of self-reported hypertension. Baseline questionnaires gathered information on any prior diagnosis of hypertension, including the age at diagnosis. Incident cases were defined as those individuals who had no history of hypertension at baseline but declared a new diagnosis during the follow-up period. All this clinical information was requested from participants in their follow-up questionnaires. Hypertension was defined following the 8th Report of the Joint National Committee criteria, which is based on repeated office measurements showing systolic blood pressure (SBP) values of ≥140 mmHg and/or diastolic blood pressure (DBP) values of ≥90 mmHg [32]. A validation study was previously conducted on a subset of this cohort to verify hypertensive status. This study confirmed the validity of self-reported diagnoses: 82.3% (95% CI: 72.8–92.8) of those reporting hypertension were confirmed through conventional blood pressure measurement, and 85.4% (95% CI: 72.4–89.1) of those who did not report hypertension were confirmed as non-hypertensive [33].

### 2.4. Covariates Assessment

The baseline questionnaire collected socio-demographic information (age, sex, marital status, educational level), validated anthropometric measures (height and weight) [27], health-related behavioral factors (smoking status, alcohol intake, physical activity [31], total energy intake (kcal/day), specific diets, adherence to the Mediterranean diet, snacking habits, sugar-sweetened beverage consumption, and mineral intake (calcium, sodium, potassium and magnesium), as well as time spent watching television. Additional information about family and personal medical history (prevalence of hypercholesterolemia and hypertriglyceridemia at baseline) and the use of analgesic drugs was also registered. The Mediterranean dietary pattern (MedDiet) was assessed through the Mediterranean diet score based on 9 items, proposed by Trichopoulou et al. [34]. Prior studies conducted within the SUN cohort have proved the reliability of self-reported data [33,35].

### 2.5. Statistical Analysis

Descriptive statistics were used to summarize baseline characteristics of the participants. Continuous variables were presented as medians and interquartile ranges (non-normal distribution) or as means and standard deviations (normal distribution), while categorical variables were expressed as proportions. Normality of distribution was verified with the Shapiro–Wilk test. The follow-up period was defined as the duration from the initial baseline questionnaire to either the date in which participants were first diagnosed with hypertension or the date of the last follow-up questionnaire, if they did not develop the outcome. To examine the association between predicted 25(OH)D status and hypertension risk, we estimated the incidence rates of hypertension across quartiles of baseline predicted serum 25(OH)D levels. We then employed Cox regression models to calculate hazard ratios and 95% CIs using the lowest quartile as the reference category.

Additionally, we investigated the continuous relationship between incident hypertension and each 1 ng/mL increase in forecasted serum 25(OH)D levels. Initially, HRs were estimated with no adjustments; afterwards, we performed different models adjusted for multiple confounders. The following model included marital status (currently married/others), cumulative tobacco exposure (smoking pack years), smoking status (never, current, or former smoker), years of university studies, daily time spent watching TV (hours), physical activity level (MET-h/week), changes in weight (defined as no change, weight gain prior baseline or prior weight loss), adherence to the Mediterranean diet (low (0–3), moderate (4–6), high (7–9)), total daily calorie intake (kcal/day), daily alcohol consumption (grams/day), adherence to a specific diet (yes/no), daily sugar-sweetened beverage intake (servings/day), intake of vitamin D supplementation (yes/no), snacking habits between meals (yes/no), and daily dietary consumption of sodium, potassium, calcium, and magnesium (mg/day). We also accounted for family history of hypertension (yes/no), prevalent hypercholesterolemia, prevalent hypertriglyceridemia, and analgesic drug use (yes/no). The third model was additionally adjusted for obesity (≥30 kg/m^2^ yes/no). Missing data were handled by simple imputation, using multivariable linear regression for continuous variables and logistic or multinomial regression for categorical variables. We imputed the following variables: daily time spent watching TV (17.8%), time spent sleeping (16.4%), following a specific diet (2.3%), sugar-sweetened beverage consumption (2.6%), weight change (2.9%), and skin reaction after sun exposure (1.6%).

We additionally assessed the potential non-linear relationship between predicted serum 25(OH)D levels and hypertension risk using restricted cubic splines. Furthermore, Nelson–Aalen curves, adjusted for potential confounding variables via the inverse probability weighting method, were performed to illustrate hypertension rates throughout the follow-up period, stratified by sufficient (>20 ng/mL) and insufficient (≤20 ng/mL) levels of predicted serum 25(OH)D. We further explored potential modifying effects by age (under or over 50 years old), sex (women/men), and overweight status (overweight/obese or normal weight). Using the likelihood ratio test, we compared the fully adjusted Cox regression model and the same model with the interaction across product terms with quartiles of estimated vitamin D (3 degrees of freedom) to assess interaction significance (*p* value for interaction). Various sensitivity analyses were conducted to estimate the fully adjusted hazard ratios after comparing the highest and the lowest quartiles of forecasted vitamin D and their association with the onset of hypertension. These analyses were conducted under various assumptions to ascertain robustness and account for potential confounders. We performed new analyses excluding extreme values of total energy intake (<1st percentile and >99th percentile), censoring participants with a follow-up period of 14 years or more, excluding those who were diagnosed with hypertension within the first 2 years of follow-up, removing outliers’ values of predicted serum 25(OH)D (±1.5 interquartile range). Further analyses excluded participants with prevalent hypertriglyceridemia and hypercholesterolemia to mitigate potential reverse causality. All *p*-values were two-tailed, and statistical significance was set at *p* < 0.05. STATA 16.0 (StataCorp, College Station, TX, USA) was used for all analyses.

## 3. Results

This study included 16,437 participants (63.8% female) with a mean age of 36.4 years (SD ± 11.1 years). The total follow-up period amounted to 193,490 person years, with a median follow-up of 12.3 years. During this time, 2338 participants developed incident hypertension. Baseline characteristics of the participants categorized by quartiles of predicted serum 25(OH)D levels are shown in Table 1. Those in the highest quartile were generally younger, predominantly unmarried, and exhibited a lower prevalence of cardiovascular risk factors such as hypercholesterolemia and hypertriglyceridemia, as well as a lower prevalence of hypertension in their family history. They were also more physically active (3.7 times more active compared to those in the first quartile), had a lower BMI, experienced more summer sun exposure (4 times greater than the lowest quartile), and used fewer analgesic drugs. Regarding dietary habits, they had a higher daily intake of vitamin D; moreover, they were less likely to follow a specific diet and snacked between meals less often than those in the first quartile. Other factors, including time watching TV or sleeping and the intake of various nutrients, minerals, and alcohol, showed no significant differences across the groups.

Most of the cases of hypertension (*n* = 842) were identified in the first quartile, with an incidence rate of 18.2 × 10^−3^ versus 8.4 × 10^−3^ years^−1^ in the highest quartile. Considering the lowest quartile as the reference, we observed a significant, monotonic inverse association between predicted serum 25(OH)D levels and the risk of hypertension throughout the whole period. This association was evident in both the unadjusted model and after adjusting for multiple variables (Table 2).

The results remained significant even after additional adjustment for obesity (BMI ≥ 30 kg/m^2^) as a potential confounder (HR_Q4vsQ1_ 0.70; 95% CI: 0.60–0.80, *p* for trend < 0.001). Furthermore, continuous analyses indicated that each 1 ng/mL increase in predicted serum 25(OH)D was linked to a 7% reduction in the risk of incident hypertension (HR 0.93; 95% CI 0.91–0.95). We conducted stratified analyses to examine whether sex, age, and obesity modified the association between predicted serum 25(OH)D and hypertension in the fully adjusted model (Table 3). No significant interactions were evinced across quartiles of predicted serum 25(OH)D for age (*p* for interaction = 0.060), sex (*p* for interaction = 0.280), or overweight status (*p* for interaction = 0.692).

We performed various sensitivity analyses to ensure the reliability of our findings. After excluding participants under different assumptions, we did not observe any significant changes in comparison to the main analysis results (Figure 2).

To address potential non-linear associations, we performed restricted cubic spline analyses adjusted for the same potential confounders, revealing a statistically significant relationship between predicted serum 25(OH)D and hypertension (*p* < 0.001). This dose–response association is graphically illustrated in Figure 3. Figure 4 exhibits the cumulative incidence of hypertension during follow-up for sufficient (>20 ng/mL) and insufficient (≤20 ng/mL) levels of predicted vitamin D, indicating a progressively higher cumulative rate of hypertension as vitamin D levels decrease.

## 4. Discussion

The current study showed that baseline levels of estimated serum 25(OH)D were inversely related to the development of hypertension over a prolonged follow-up period. The association remained statistically significant after adjusting for multiple potential confounders. The risk of hypertension was also independently associated with predicted 25(OH)D levels after adjusting for obesity, considering the cut-off point of 30 kg/m^2^. BMI was not adjusted as a continuous variable since it was considered a potential intermediate factor in the causal pathway. Results indicated a 30% lower risk of hypertension in the highest quartile compared to the lowest quartile of predicted vitamin D levels (HR 0.70; 95% CI: 0.60–0.80, *p* trend < 0.001). Additionally, a 7% decrease in hypertension incidence was observed for each 1 ng/mL increase in predicted vitamin D levels (HR 0.93; 95% CI: 0.91–0.95). These findings continued to be significant after performing several sensitivity analyses.

Our findings agree with previously published evidence. A comprehensive meta-analysis identified a significant inverse association between incident arterial hypertension and baseline serum levels of 25(OH)D, revealing a relative risk for high blood pressure of 0.70 (95% CI: 0.57–0.86) [18]. Another updated meta-analysis of cohort studies and randomized trials suggested that the risk of new hypertension events decreased by 7% (RR 0.93; 95% CI: 0.89–0.98), after each increase of 10 ng/mL [22]. In addition, a clinical trial conducted among the Black population showed a significant reduction of 0.2 mmHg in systolic pressure per 1 ng/mL increase in serum 25(OH)D (*p* = 0.02), although there was no impact on diastolic pressure [36]. Furthermore, another trial found that vitamin D deficient individuals (<15 ng/mL) had a relative risk of developing hypertension of 2.67 (95% CI: 1.05–6.79) in women and 6.13 (95% CI: 1.00–37.8) in men after four years of follow-up [37]. Analyses of Mendelian randomization supported the causal positive effect of increased serum 25(OH)D in lowering the risk of hypertension [38,39]. One of the analyses documented a reduction in SBP by 0.37 mmHg (95% CI: −0.73 to 0.003; *p* = 0.052) and 0.29 mmHg (95% CI: −0.52 to −0.07; *p* = 0.01) in DBP after each increase of 10% in genetically determined vitamin D levels [38]. Previous evidence has suggested the existence of baseline thresholds for serum vitamin D levels, whereas some researchers evinced a considerably higher risk of hypertension in those participants with 25(OH)D below 30 ng/mL that remained statistically significant over the range of 30–52 ng/mL [22], while others considered a threshold of 23 ng/mL [40].

Meanwhile, most interventional studies have yielded limited evidence supporting the beneficial impact of vitamin D supplementation on hypertension [14,40,41,42,43,44]. A meta-analysis of 46 clinical trials reported that among participants with a mean baseline vitamin D below 20 ng/mL supplementation did not lead to significant changes in blood pressure [21]. Based on this finding, they concluded that vitamin D supplementation might not be an effective strategy for lowering blood pressure [21]. Similarly, He S et al. [20] conducted another meta-analysis and found no effect on blood pressure among individuals with deficient levels of vitamin D. Nonetheless, their subgroup analyses revealed a significant improvement in blood pressure in both obese and participants over 50 years old after vitamin D3 supplementation [20]. Similarly, Golzarand et al. [19] found positive effects in groups that received daily doses over 800 IU during less than 6 months, as well as in older participants. These discrepancies could be explained by the high heterogeneity of the recruited populations. Therefore, in this study, we limited our recruitment to those apparently healthy participants, leaving aside targeted diseases (such as diabetes, cardiovascular events, or cancer), which could have interfered in the interaction between vitamin D and hypertension. Moreover, differences in dosing regimens can indeed influence the outcomes shown in different studies. On the one hand, studies with more frequent dosing schedules (daily or weekly doses) might better mimic the physiological vitamin D synthesis and provide more consistent benefits for blood pressure regulation [40]. On the other hand, monthly high-dose vitamin D3 supplementation results in large spikes in 25(OH)D status, followed by a gradual decline over the course of the month, which could lead to periods of suboptimal vitamin D levels in-between doses and might potentially negate any consistent effects on hypertension [42].

There are several mainstream mechanisms that can explain how vitamin deficiency may induce hypertension. The first mechanism involves the activation of the renin-angiotensin-aldosterone system (RAAS). As the plasma renin concentration increases, sympathetic activity might become intensified and consequently enhance intra-glomerular pressure, leading to higher blood pressure, a lower glomerular filtration rate (GFR), and cardiovascular damage [13,20,23,45]. Secondly, vitamin D is involved in calcium homeostasis as it stimulates calcium transporters, increasing renal calcium reabsorption and its release from bones. As a consequence, deficient levels of serum vitamin D might result in a lower plasma concentration of calcium, leading to a higher secretion of the parathyroid hormone (PTH), which has proven to be inversely correlated with hypertension [13]. Although the mechanism is still unclear, it is hypothesized that PTH increases the levels of calcium in blood stream, leading to endothelial dysfunction [13]. Moreover, vitamin D may be directly implicated in left ventricular hypertrophy and vascular stiffness [9,46]. Insufficient levels of serum 25(OH)D might lead to a decreased NO (nitric oxide) in blood vessels and a reduction in the calcium influx, affecting vasodilation and causing a subsequent increase in blood pressure [20].

Although evidence does not support vitamin D’s antihypertensive therapeutic effect alone, studies examining vitamin D’s role as an adjuvant agent to the classical antihypertensive treatment in patients diagnosed with hypertension and vitamin D deficiency seemed to have promising results [20,23,47,48]. If this beneficial effect is confirmed in further research, preventing vitamin D deficiency through food supplementation may be considered as a cost-effective measure to prevent and control hypertension, rather than a widespread screening of vitamin D deficiency as its high prevalence has been clearly demonstrated worldwide [49,50].

### Limitations and Strengths

Our study has some limitations that must be noted. The sample primarily consists of young adults with a high level of education, making it unrepresentative of the general population. Consequently, generalizing our findings requires a deeper understanding of the specific conditions and shared biological mechanisms, rather than relying solely on statistical representativeness [51]. However, this characteristic of our study also enhances its internal validity due to the participants’ high educational level and homogeneity, which minimize the influence of confounding factors related to education and their socioeconomic status. Additionally, the use of self-reported data (a validated FFQ to evaluate dietary pattern and lifestyle habits) introduces subjectivity and potential information bias. This is particularly relevant when reporting daily consumption of vitamin D in both diet and supplements, given that collected data consistently indicate consumption below recommended levels. Nevertheless, the prior validation of these self-reported parameters helps ensure the reliability of our results [33,35]. We did not have specific information about issues such as the type of sunscreen used or the clothing, which could be considered in further research. Moreover, our vitamin D prediction model is primarily a research tool more suitable for epidemiological contexts rather than daily clinical practice. It may not be appropriate for specific groups, such as children, pregnant women, or individuals with severe kidney disease, who require tailored prediction methods for accurate outcomes.

To our knowledge, this is one of the few studies to assess the association between forecasted serum 25(OH)D and incident hypertension in a Spanish population and the first one based on a validated predictive model. The study design offers several strengths, including its dynamic recruitment strategy, prospective nature, large sample size with substantial long-term follow-up, and a high retention rate. These factors are essential for establishing a clear cause-and-effect relationship between vitamin D and hypertension. We also accounted for multiple potential confounders and conducted several sensitivity analyses to strengthen the reliability of our findings, minimizing the likelihood of residual bias.

## 5. Conclusions

Baseline levels of predicted serum 25(OH)D, based on a validated model, appeared to show an independent and inverse association with an increased risk of developing hypertension in a Mediterranean cohort. The association remained strongly significant even after adjusting for multiple confounders, implying that higher levels of forecasted vitamin D at the outset may offer protective benefits against incident hypertension. Nonetheless, it remains uncertain what specific baseline threshold of 25(OH)D might be considered preventive for hypertension, and further investigations should be performed.

## Figures and Tables

**Figure 1 nutrients-16-02351-f001:**
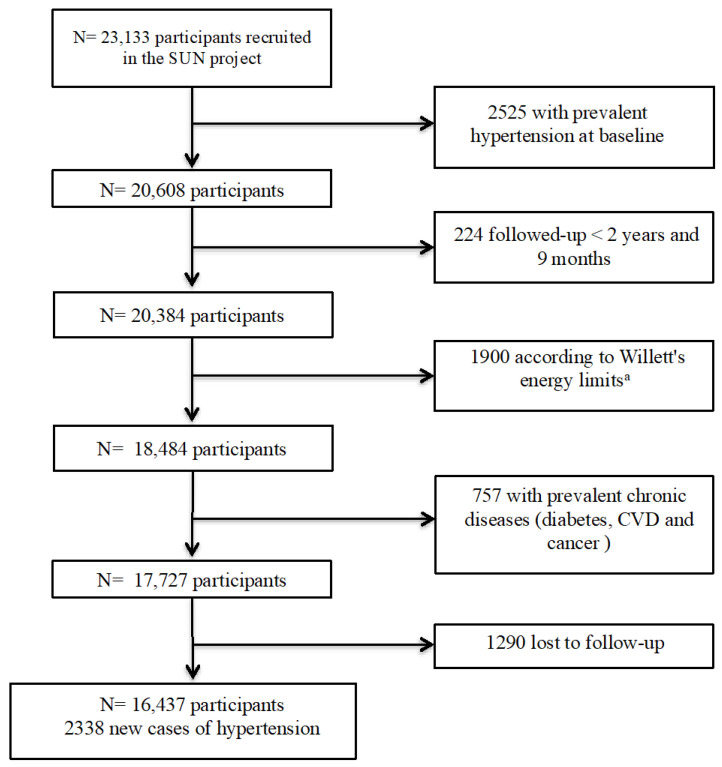
Flowchart of participants included in the analyses. ^a^ Energy limits predefined by Willett (women: <500 kcal/d or >3500 kcal/d; men: <800 kcal/d or >4000 kcal/d).

**Figure 2 nutrients-16-02351-f002:**
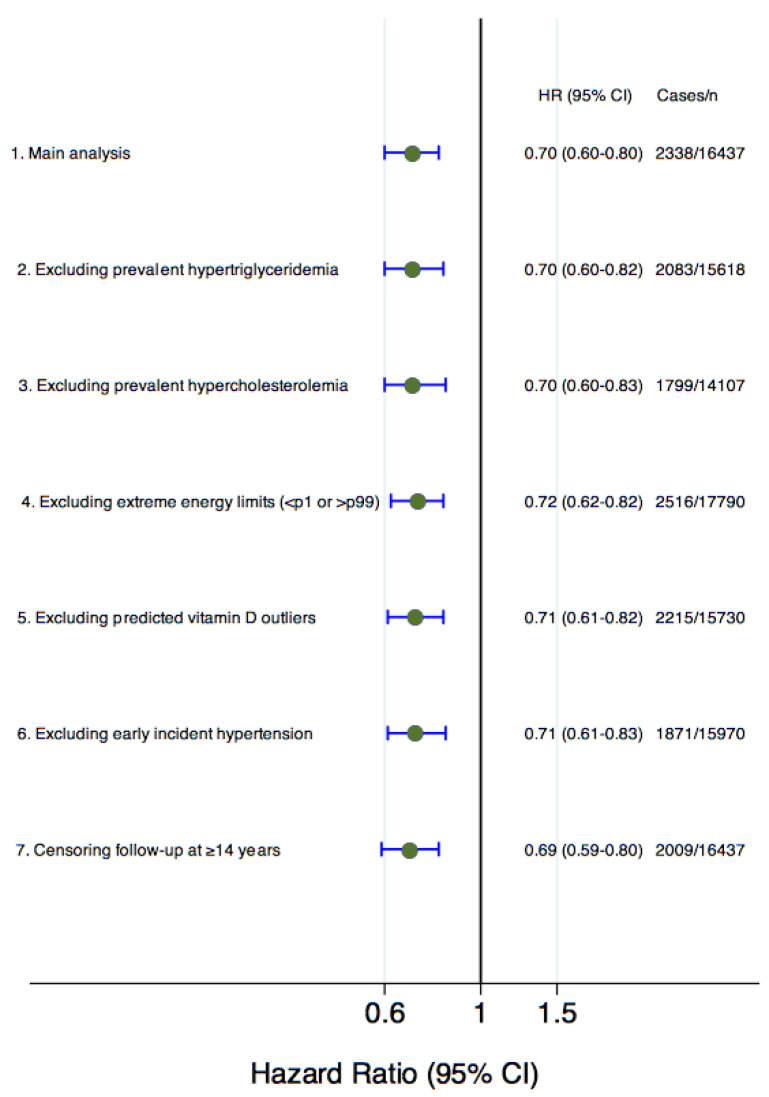
Sensitivity analyses: HR and 95% CI (Q4 vs. Q1) for incident hypertension according to forecasted serum 25(OH)D status (comparison of upper versus lowest quartile). 1. Fully adjusted model for age, sex, marital status, lifetime smoking (pack years), smoking status, years at university, daily TV consumption, physical activity, weight change, adherence to Mediterranean diet score, energy intake, alcohol consumption, following a specific diet, sugar-sweetened beverage intake, vitamin D supplementation, snacking, dietary consumption of sodium, potassium, calcium, and magnesium, family history of hypertension, prevalent hypercholesterolemia, prevalent hypertriglyceridemia, use of analgesic drugs, and obesity. 2. Without prevalent hypertriglyceridemia adjustment. 3. Without prevalent hypercholesterolemia adjustment. 4. Without extreme daily energy intake (<p1 or >p99). 5. Without outliers located within ±1.5 interquartile range of the average of predicted vitamin D. 6. Without participants who were diagnosed of hypertension during the first 2 years of follow-up. 7. Without participants with a follow-up period of 14 years or more.

**Figure 3 nutrients-16-02351-f003:**
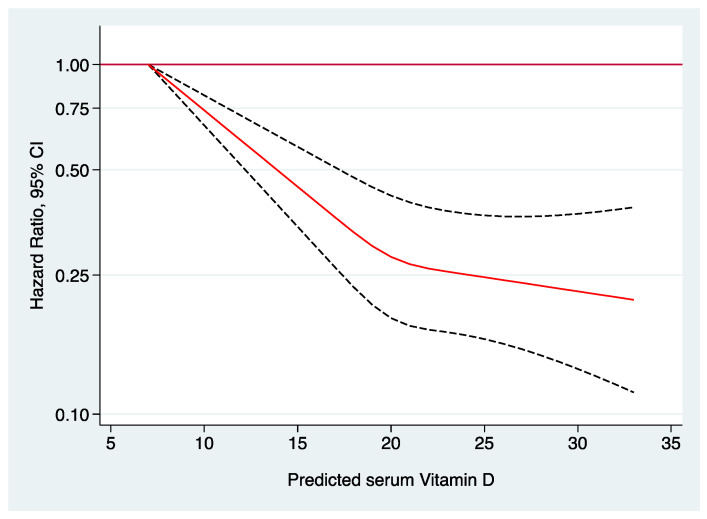
Restricted cubic splines dose–response association: Adjusted HR * and 95% CI for the risk of developing hypertension according to predicted levels of serum 25(OH)D. * Adjusted for age, sex, marital status, lifetime smoking (pack years), smoking status, years at university, daily TV consumption, physical activity, weight change, adherence to Mediterranean diet score, energy intake, alcohol consumption, following a specific diet, sugar-sweetened beverage intake, vitamin D supplementation, snacking, dietary consumption of sodium, potassium, calcium, and magnesium, family history of hypertension, prevalent hypercholesterolemia, prevalent hypertriglyceridemia, use of analgesic drugs, and obesity. Red line represents HR, and dotted lines represent 95% confidence intervals.

**Figure 4 nutrients-16-02351-f004:**
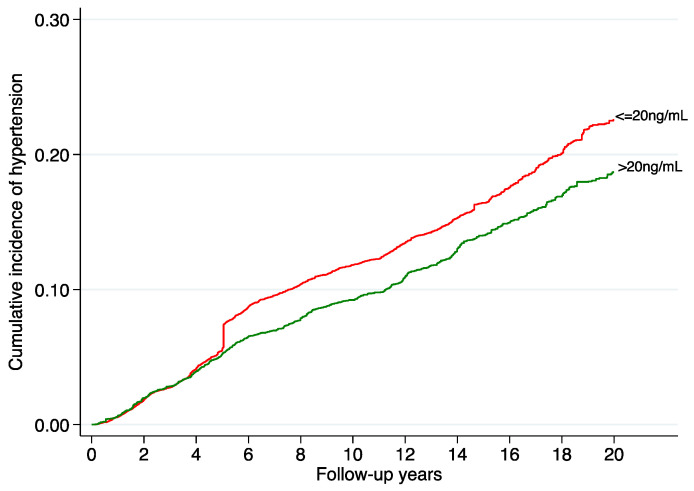
Nelson–Aalen graph: cumulative incidence of hypertension during follow-up according to predicted vitamin D status (cut-off point of 20 ng/mL). Adjusted for age, sex, marital status, lifetime smoking (pack years), smoking status, years at university, daily TV consumption, physical activity, weight change, adherence to Mediterranean diet score, energy intake, alcohol consumption, following a specific diet, sugar-sweetened beverage intake, vitamin D supplementation, snacking, dietary consumption of sodium, potassium, calcium, and magnesium, family history of hypertension, prevalent hypercholesterolemia, prevalent hypertriglyceridemia, use of analgesic drugs, and obesity.

**Table 1 nutrients-16-02351-t001:** Baseline features according to quartiles of predicted serum 25(OH)D.

	Quartiles of Predicted Serum 25(OH)D			
	Q1	Q2	Q3	Q4
N	4110	4109	4109	4109
Serum 25(OH)D predicted status (range, ng/mL)	7.2; 18.7	18.8; 19.8	19.9; 21.3	21.4; 32.7
Age (years) (p25; p75)	38.8 (31.0; 47.5)	36.0 (29.0; 45.0)	32.3 (26.0; 42.0)	30.0 (24.8; 39.8)
Women (%)	66.1	64.1	63.9	61.0
Smoking status (%)				
Never	45.7	47.8	52.6	56.1
Current	21.9	23.3	23.1	22.0
Former	32.4	28.9	24.3	21.9
Marital status, married (%)	58.1	52.8	42.5	35.6
Years of university (years) (p25; p75)	5.0 (4.0; 5.0)	5.0 (4.0; 5.0)	5.0 (4.0; 5.0)	5.0 (4.0; 5.0)
Body Mass Index (kg/m^2^) (p25; p75)	25.2 (22.9; 27.8)	23.1 (21.1; 25.0)	21.9 (20.2; 23.9)	21.5 (19.8; 23.5)
Weight change (%) ^1^				
No weight change	21.8	26.3	30.6	30.5
Weight gain	58.6	51.1	43.1	37.9
Weight loss	19.6	22.6	26.3	31.6
Physical activity (METs-h/wk) (p25; p75)	6.9 (1.6; 15.6)	12.4 (4.5; 21.9)	19.2 (9.1; 31.2)	33.4 (18.2; 54.1)
TV (hours/day) (p25; p75) *	1.5 (0.8; 2.0)	1.4 (0.8; 2.0)	1.5 (0.8; 2.0)	1.5 (0.8; 2.0)
Siesta (hours/day) (SD)	0.6 (0.5)	0.5 (0.5)	0.5 (0.5)	0.5 (0.5)
Sleeping hours (hours/day) (p25; p75) *	7.3 (7.0; 8.0)	7.3 (7.0; 8.0)	7.3 (7.0; 8.0)	7.3 (7.0; 8.0)
Walking time (min/day) (p25; p75)	15.0 (15.0; 25.0)	25.0 (15.0; 25.0)	25.0 (15.0; 45.0)	45.0 (25.0; 90.0)
Summer sun exposure (h/day) (p25; p75)	0.4 (0.1; 0.9)	0.8 (0.3; 1.2)	1.0 (0.5; 1.6)	1.6 (0.8; 3.0)
Skin reaction after sun exposure (%) *				
Mild reaction	74.6	98.3	99.1	99.4
Severe reaction	25.4	1.7	0.9	0.6
Energy intake (kcal/d) (p25; p75)	2263.2 (1851.0; 2715.1)	2268.0 (1869.8; 2712.1)	2346.1 (1941.1; 2786.1)	2436.0 (2005.4; 2901.7)
Carbohydrate intake (% of energy) (p25; p75)	43.0 (38.4; 47.8)	43.6 (38.7; 48.1)	43.8 (39.2; 48.2)	43.8 (39.2; 48.4)
Protein intake (% of energy) (p25; p75)	18.1 (16.2; 20.4)	17.9 (16.0; 19.9)	17.8 (16.0; 20.0)	18.0 (16.2; 20.1)
Fat intake (% of energy) (p25; p75)	36.8 (32.4; 40.6)	36.3 (32.6; 40.7)	36.3 (32.2; 40.5)	36.2 (32.0; 40.2)
Monounsaturated fatty acids intake (% of energy) (p25; p75)	15.6 (13.7; 18.1)	15.6 (13.5; 17.9)	15.5 (13.4; 17.8)	15.3 (13.3; 17.6)
Saturated fatty acids intake (% of energy) (p25; p75)	12.4 (10.5; 14.5)	12.5 (10.7; 14.5)	12.5 (10.6; 14.5)	12.4 (10.3; 14.4)
Polyunsaturated fatty acids intake (% of energy) (p25; p75)	5.0 (4.1; 6.0)	5.0 (4.2; 6.0)	5.0 (4.2; 6.1)	5.0 (4.1; 6.1)
Meat consumption (g/d) (p25; p75)	169.2 (122.4; 221.7)	167.4 (121.6; 219.4)	170.7 (124.3; 221.0)	171.6 (123.6; 227.2)
Trichopoulou’s Mediterranean diet score (p25; p75)	4.0 (3.0; 5.0)	4.0 (3.0; 5.0)	4.0 (3.0; 5.0)	4.0 (3.0; 6.0)
Sodium intake (mg/d) (p25; p75)	2756 (2064; 3733)	2799 (2077; 3785)	2862 (2148; 3861)	2963 (2191; 3945)
Potassium intake (mg/d) (p25; p75)	4482 (3622; 5468)	4434 (3616; 5393)	4584 (3743; 5601)	4818 (3883; 5937)
Calcium intake (mg/d) (p25; p75)	1145 (886; 1444)	1132 (881; 1417)	1173 (921; 1482)	1231 (953; 1552)
Magnesium intake (mg/d) (p25; p75)	393 (321; 479)	389 (322; 472)	404 (335; 486)	423 (347; 512)
Following specific diet (%) *	10.4	7.2	6.3	7.5
Between-meal snacking (%)	38.1	31.9	32.8	34.6
Alcohol intake (g/d) (p25; p75)	2.3 (0.6; 8.1)	2.8 (1.0; 8.6)	3.2 (1.0; 8.5)	3.2 (1.0; 8.5)
Sugar-sweetened beverage (servings/d) (p25; p75) ^2^*	0.1 (0.0; 0.1)	0.1 (0.0; 0.1)	0.1 (0.0; 0.4)	0.1 (0.0; 0.4)
Total vitamin D intake (mcg/d) ^3^ (p25; p75)	4.8 (3.5; 6.9)	4.9 (3.6; 7.6)	5.2 (3.8; 10.5)	5.6 (3.9; 11.0)
Vitamin D supplementation (%)	6.4	7.8	9.3	10.9
Vitamin D supplementation (mcg/d) ^4^ (p25; p75)	2.1 (0.3; 5.0)	2.1 (0.7; 5.0)	3.9 (0.7; 5.0)	5.0 (0.7; 5.0)
Total dietary fiber intake (g/d) (p25; p75)	20.3 (15.3; 26.4)	20.0 (15.3; 26.1)	20.8 (16.1; 26.8)	21.9 (16.7; 28.6)
Caffeine intake (mg/d) (p25; p75)	5.0 (2.0; 6.0)	5.0 (2.0; 6.0)	5.0 (1.0; 6.0)	5.0 (1.0; 6.0)
Use of analgesic drugs (%)	12.5	11.2	9.2	9.3
Family History of Hypertension (%)	45.1	41.9	38.0	36.4
Prevalent Hypercholesterolemia (%)	17.9	14.9	12.3	11.6
Prevalent Hypertriglyceridemia (%)	7.4	5.2	3.8	3.6

Categorical variables are described as percentages and continuous variables as medians (interquartile range), except for siesta, which shows a normal distribution and therefore is described as mean (standard deviation). ^1^ Weight change prior baseline. ^2^ One serving of sugar-sweetened beverages = 200 mL. ^3^ Total intake of vitamin D and supplementation, energy-adjusted by residual method (mcg/d). ^4^ Median intake of vitamin D supplementation (mcg/day) was calculated among consumers of vitamin supplements. * Imputed values: watching TV (17.8%), time spent sleeping (16.4%), following a specific diet (2.3%), sugar-sweetened beverages consumption (2.6%), weight change (2.9%) and skin reaction after sun exposure (1.6%).

**Table 2 nutrients-16-02351-t002:** Cox proportional HRs and 95% CI for incident HT according to quartiles of predicted 25(OH)D status at baseline.

	Q1	Q2	Q3	Q4	*p* Trend	Continuous per + 1 ng/mL ^d^
Predicted serum 25(OH)D ^a^	17.75 (16.63; 18.33)	19.35 (19.05; 19.63)	20.52 (20.20; 20.87)	22.48 (21.78; 23.59)		
Cases of Incident HT	842	637	447	412		
Person years	46,144	48,251	50,020	49,075		
Incident rate 10^−3^ years^−1^	18.2	13.2	8.9	8.4		
Age and sex adjusted model	1.00 (reference)	0.79 (0.71–0.87)	0.59 (0.52–0.66)	0.58 (0.52–0.66)	<0.001	0.90 (0.88–0.92)
Multiple-adjusted model1 ^b^	1.00 (reference)	0.86 (0.77–0.95)	0.66 (0.58–0.75)	0.67 (0.58–0.77)	<0.001	0.92 (0.90–0.94)
Multiple-adjusted model2 ^c^	1.00 (reference)	0.88 (0.79–0.98)	0.68 (0.60–0.77)	0.70 (0.60–0.80)	<0.001	0.93 (0.91–0.95)

^a^ Predicted serum 25(OH)D status (ng/mL) expressed by p50 (p25; p75). ^b^ Model adjusted for age, sex, marital status, lifetime smoking (pack years), smoking status (never, current or former), years at university, daily TV consumption, physical activity (MET-h/week), weight change, adherence to Mediterranean diet score, energy intake (kcal/day), alcohol consumption (g/day), following a specific diet, sugar-sweetened beverage intake (servings/day), vitamin D supplementation (%), snacking (%), dietary consumption of sodium, potassium, calcium, and magnesium (mg/day), family history of hypertension, prevalent hypercholesterolemia, prevalent hypertriglyceridemia, and use of analgesic drugs. ^c^ Model 1 additionally adjusted for obesity. (≥30 kg/m^2^ yes, no). ^d^ HR and 95% CI for incident hypertension for each 1ng/mL increase in predicted serum 25(OH)D (*p* < 0.005).

**Table 3 nutrients-16-02351-t003:** Analysis of effect modification: Adjusted HRs * and 95% CI for the risk of hypertension according to quartiles of predicted levels of serum 25(OH)D stratified by potential confounders.

	N	Incident HT	Q1	Q2	Q3	Q4	*p* for Interaction
Age							0.060
Age ≥ 50 years	2103	703	1.00 (reference)	0.87 (0.71–1.06)	0.85 (0.67–1.06)	0.79 (0.61–1.05)	
Age < 50 years	14,334	1635	1.00 (reference)	0.88 (0.77–0.99)	0.61 (0.53–0.71)	0.64 (0.54–0.76)	
Sex							0.280
Women	10,483	992	1.00 (reference)	0.83 (0.70–0.97)	0.66 (0.54–0.80)	0.75 (0.61–0.93)	
Men	5954	1346	1.00 (reference)	0.94 (0.81–1.09)	0.72 (0.61–0.85)	0.68 (0.56–0.82)	
Overweight							0.692
Overweight/Obese	4267	1155	1.00 (reference)	1.01 (0.87–1.17)	0.84 (0.69–1.03)	0.80 (0.62–1.04)	
Normal weight	12,170	1183	1.00 (reference)	0.93 (0.78–1.10)	0.78 (0.65–0.93)	0.86 (0.70–1.05)	

* Adjusted for age, sex, marital status, lifetime smoking (pack years), smoking status, years at university, daily TV consumption, physical activity, weight change, adherence to Mediterranean diet score, energy intake, alcohol consumption, following a specific diet, sugar-sweetened beverage intake, vitamin D supplementation, snacking, dietary consumption of sodium, potassium, calcium, and magnesium, family history of hypertension, prevalent hypercholesterolemia, prevalent hypertriglyceridemia, use of analgesic drugs, and obesity.

## Data Availability

The data from the SUN Project that support our findings are available upon request from the Department of Preventive Medicine and Public Health, School of Medicine, University of Navarra (Spain) at sun@unav.es.

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
