# Peer review of "Vitamin D and the Risk of Developing Hypertension in the SUN Project: A Prospective Cohort Study"

_nutrients, 2024, doi:10.3390/nu16142351_

Round 1

Reviewer 1 Report

Comments and Suggestions for Authors

The manuscript entitled “Vitamin D and risk of developing hypertension in the SUN project: A prospective cohort study” addresses an important topic and is quire well-written.

There are some methodological challenges in the study that are hard to avoid given the project's assumptions and the study population, but it is crucial to describe them well and then discuss the limitations.

      Vitamin D (25(OH)D) is found in various dietary sources, and it is also synthesized endogenously through skin sun exposure [4]. - Please be precise in your wording. The real dietary sources of vitamin D are fatty fish, specifically certain species like salmon, rainbow trout, and herring. Other dietary sources are almost negligible, of course, excluding fortified products.

      Lines 105-106 – “Vitamin D intake from both dietary sources and supplements was consolidated into a single variable,” – I would approach combining these two sources of vitamin D into a single variable with great caution. The absorption and metabolism of vitamin D are very complex and also depend on its release from the food matrix and the presence (or absence) of fat in supplements. Were analyses considering the separation of these data taken into account?

      Lines 107-108 – “Summer sun exposure was assessed after calculating a weighted average of the exposure to sunlight each week, during summer months” – Did the calculations (or questions) include information about the use of sunscreen or sun avoidance? Such a calculation is subject to a very high margin of error.

      Lines 135-136 – “and mineral intake (calcium, sodium, potassium and magnesium)” – Why only these minerals? What about vitamins?

      Lines 145- 146 – “Continuous variables were presented as means and standard deviations, while categorical variables were expressed as proportions” – Was the normality of distribution tested? The information about it should be added and authors should be consequent. If data have normal distribution, they should be treated as such, if not, nonparametric tests should be applied.  Please specify it.

      Line 168-170 – “Missing data were handled by simple imputation, using multivariable linear regression for continuous variables and logistic or multinomial regression for categorical variables.” - What was the scale of missing data, and in which categories? Please indicate this in the table as well.

      Table 1 – “Weight change” - Considering that vitamin D is fat-soluble and accumulates in adipose tissue, please add a variable related to weight loss (if such data is available). If there are cases of weight loss, some stored vitamin D may be released.

      Considering the fact that some data are self-reported while others are calculated/predicted (I also pointed out potential errors that could affect these calculations), the authors should particularly detail the study limitations (and discuss them in the context of the literature). Please separate the section entitled “limitations of the study”.

      The conclusions in the paper are better formulated than in the abstract - perhaps it would be worth considering adding them to the abstract in this version

Reviewer 2 Report

Comments and Suggestions for Authors

This is an interesting study that contributes to knowledge of vitamin D and hypertension. There are many strengths, such as the large size of the cohort, the long follow-up, and the long list of potential confounders that were measured and used for adjustment.  The Discussion is very thoughtful and outs the findings of these researchers and others into perspective.

There are some comments the authors need to address.

1.        Please be consistent and precise in using terms “vitamin D” and “25(OH)D”.

a.         Line 46 one does not have 25(OH)D as the abbreviation for vitamin D. The former refers only to the transport form while “vitamin D” is both the parent compound and a general term for vitamin D activity.  When referring to the transport form which is used for status always call it 25(OH)D

b.       Line 46-47 implies that 25(OH)D is found in dietary sources which may be true in some situations (e.g., meat from animals fed calcidiol) however most of our food contain the parent compounds cholecalciferol and ergocalciferol.

c.        Line 48-49. It is not 25(OH)D directly that has activity it is the active form 1,25(OH)2D.

d.       In Methods the term “serum vitamin D” is used when it should be "predicted serum 25(OH)D" levels.

2.        The Introduction is somewhat weak in providing a rationale for the study (lines 51-62) yet the Discussion especially lines 381-400 is very strong in showing the current cohort evidence. Since one might assume Spain has sufficient sun exposure for vitamin D synthesis the Introduction should be clearer why this study was done.

3.        A strength of the study is testing for an obesity effect . Q1 has mean BMI of 25.5 while Q4’s BMI mean is 21.4 which is a considerable difference.  Here could be a reason for difference in blood pressure. Also consider the effect obesity has on 25(OH)D levels- most studies show it is lower with obesity.

4.        The RCTs not showing a treatment effect may themselves have reasons for a lack of effect. Reference 42 is the study led by Scragg and in his RCT, vitamin D was given monthly, a regimen that has been criticized as not being physiological  (with daily and weekly better at mimicking sun exposure). You might comment on that point.
